# A Head/Tail Breaks-Based Approach to Characterizing Space-Time Risks of COVID-19 Epidemic in China's Cities

**Tingting Wu [1], Bisong Hu [1,*] , Jin Luo [1] and Shuhua Qi [1,2]**

[1] School of Geography and Environment, Jiangxi Normal University, Nanchang 330022, China; 202140100105@jxnu.edu.cn (T.W.); luojin@jxnu.edu.cn (J.L.); qishuhua11@jxnu.edu.cn (S.Q.)

[2] Key Laboratory of Poyang Lake Wetland and Watershed Research, Ministry of Education, Jiangxi Normal University, Nanchang 330022, China

[*] Correspondence: hubisong@jxnu.edu.cn

**Abstract:** The novel coronavirus pneumonia (COVID-19) pandemic has caused enormous impacts around the world. Characterizing the risk dynamics for urgent epidemics such as COVID-19 is of great benefit to epidemic control and emergency management. This article presents a novel approach to characterizing the space-time risks of the COVID-19 epidemic. We analyzed the heavy-tailed distribution and spatial hierarchy of confirmed COVID-19 cases in 367 cities from 20 January to 12 April 2020, and population density data for 2019, and modelled two parameters, COVID-19 confirmed cases and population density, to measure the risk value of each city and assess the epidemic from the perspective of spatial and temporal changes. The evolution pattern of high-risk areas was assessed from a spatial and temporal perspective. The number of high-risk cities decreased from 57 in week 1 to 6 in week 12. The results show that the risk measurement model based on the head/tail breaks approach can describe the spatial and temporal evolution characteristics of the risk of COVID-19, and can better predict the risk trend of future epidemics in each city and identify the risk of future epidemics even during low incidence periods. Compared with the traditional risk assessment method model, it pays more attention to the differences in the spatial level of each city and provides a new perspective for the assessment of the risk level of epidemic transmission. It has generality and flexibility and provides a certain reference for the prevention of infectious diseases as well as a theoretical basis for government implementation strategies.

**Keywords:** COVID-19; risk assessment; head/tail breaks; power law; ht index

## 1. Introduction

The COVID-19 pandemic caused by the novel coronavirus (SARS-CoV-2) infection has constituted a global health security threat since 2020 [1–3]. As of June 2023, the COVID-19 pandemic has lasted over three years, posing great challenges to regional epidemic prevention and control [4]. Characterizing the risk dynamics of the COVID-19 epidemic is of great benefit to epidemic control and emergency management. It is urgent to establish an effective assessment system to identify space-time risks which can provide support for formulating epidemic control and prevention policies [5]. Since the COVID-19 outbreak, governments have launched various measures to combat the pandemic, such as quarantines, the closure of cities, and the suspension of large gatherings [6,7]. Meanwhile, researchers have conducted a large number of studies to combat the COVID-19 pandemic, focusing on the clinical characteristics [8], viral gene sequencing [9], epidemic transmission characteristics [10,11], spatiotemporal distribution patterns [12–15], space-time risk assessments [16,17], and others.

With the advent of the post-epidemic era, the study of space-time risk assessment of the COVID-19 epidemic has gradually become a focus of attention [18,19]. Various data sources related to the epidemic's spread, e.g., social and environmental data, were used to

identify the dynamics of epidemic risks. The population-related data played an important role in risk assessment, such as population movement data [20,21], residential population data [22], and population density data [23], which were usually correlated with reported cases to support risk modelling. In addition, spatiotemporal big data such as Tencent location-based big data can be applied to indicate the population flow and help construct an epidemic model to evaluate the risk of the epidemic [24]. Smartphone signaling data can also be used to indicate human mobility and support modelling the human-to-human transmission networks to assess the risk of COVID-19 infections [25]. Many scholars have assessed the risks of COVID-19 transmission in different countries and regions associated with various natural and social environmental factors [16,25], and results have shown that the COVID-19 epidemic spread was due to multiple environmental, economic, and social factors [26], in which population density was an important determinant of the infection.

On the other hand, many researchers have proposed corresponding modelling approaches for the spatiotemporal spread of COVID-19 epidemics. The most common approach is a series of models derived from the classical SIR model [27,28] based on the principles of infectious disease dynamics, including SEIR models [29–31], SEAIR models [32,33], etc., which simulate the COVID-19 epidemic situation in cities. Key epidemiological parameters such as the basic reproduction number ($R_0$) [34,35] have found widespread application in monitoring the transmission dynamics and risk assessment of COVID-19. Additionally, scholars have applied various methods such as spatial stratified heterogeneity statistics [34], Kalman filtering [36–38], Bayesian maximum entropy [39], head/tail breaks [40], and more [41–45]. These methods have been employed for various applications in modelling the spatiotemporal spread of COVID-19, including identifying spatiotemporal distribution patterns [46,47], mapping spatiotemporal disease distributions [48], describing spatiotemporal characteristics of the epidemic [49–51], and assessing its impact on production and daily life in China [52–54]. These studies cover different stages of epidemic transmission.

China, with its vast territory and large population, exhibits significant regional disparities. Risk assessment and early warning require hierarchical zoning, with different regions at various risk levels adopting different prevention and control measures. However, research in this area remains limited. The factors influencing the transmission and spread of the COVID-19 virus are multifaceted and include not only the intrinsic biological factors of the infected [55–57] but also natural and social factors [47,58] such as temperature [58], population mobility [59], and economic development [60,61]. Relevant studies have indicated that the spatial hierarchy of epidemic cases is correlated with the spatial hierarchy of the underlying population. Comparing the spatial hierarchies of COVID-19 cases and the population, the results show that the pandemic is largely determined by the underlying population, with an $R^2$ value as high as 0.82 [40]. In simple terms, the more the population, the more cases there are. For a specific study area, the cases and the population are distributed across different units, such as different cities, forming different spatial hierarchies. Therefore, we can identify the potential risk of epidemic transmission by comparing the differences in their relative hierarchies.

With this consideration in mind, our research endeavors to introduce a novel approach to characterizing the space-time risk of COVID-19 outbreaks in Chinese cities. We employ a "head/tail breaks approach" [62–64] to delineate the spatial hierarchy of the COVID-19 cases and populations within each city for every date or specified period (e.g., weekly). Subsequently, we compare the disparities in their relative hierarchies. We then define a risk assessment metric to ascertain the risk levels for each city. This approach enables the examination of the dynamic risk of epidemic spread in both the spatial and temporal domains. Unlike traditional classification and stratification methods, the head/tail breaks method applies to data with heavy tail distribution [62,65] and can better reveal the inherent hierarchical structures of things, which can effectively portray regional differences in the space-time risk of the COVID-19 epidemic. At present, although there have been studies applying the head/tail breaks approach to the space-time mapping of the COVID-19

epidemic [40], there are few, if any, studies on the assessment of the risk level of COVID-19 epidemics. Our study provides a direct method for describing the space-time risk of epidemic transmission, offering a fresh perspective on the dynamics of epidemic risk assessment. Importantly, the methodology we propose not only elucidates the inherent hierarchy of epidemic cases and populations at a macroscopic level but also allows for the characterization of risk dynamics at a local level, such as within specific regions or individual cities. This provides a theoretical basis for government policy implementation, offers insights for domestic and international efforts in the prevention and control of COVID-19, and imparts valuable lessons for enhancing the utilization of healthcare services in other regions.

## 2. Data and Methodology

### 2.1. Study Area and Data Sources

This paper, using China as a case study, examines the space-time evolution of COVID-19 risk. To account for the typicality of the COVID-19 pandemic, this study acquired daily confirmed COVID-19 cases for 367 municipal-level administrative units in China from 20 January 2020 to 12 April 2020, from sources including the National Health Commission of the People's Republic of China (www.nhc.gov.cn, accessed on 1 April 2020) and official websites of municipal health commissions. As of 12 April, China had reported 108 new confirmed cases, bringing the total cumulative confirmed cases to 82,160. The development of the COVID-19 situation in China exhibits distinct phases. Based on the trend in daily new confirmed cases (as shown in Figure 1a), the pandemic in China can be divided into three stages: a rapid growth phase (20 January–12 February), a controlled reduction phase (13 February–1 March), and a stable maintenance phase (2 March–12 April). It is noteworthy that, on 12 February, there was a peak in the number of new confirmed cases due to a change in the statistical criteria, which included clinical diagnosis cases as confirmed cases. This change aided in controlling the outbreak, increasing the recovery rate, and reducing the mortality rate. Spatially (as depicted in Figure 1b), the confirmed cases are concentrated in the eastern regions, particularly in areas surrounding Wuhan, such as Huanggang, Xiaogan, Chongqing, and other cities, which were significantly affected by the pandemic. In contrast, the northwestern and southwestern regions experienced a relatively lower impact, with fewer new cases of COVID-19. The spread of the pandemic exhibits periodic characteristics, and we will explore the space-time evolution of the COVID-19 risk in various Chinese cities over 12 weeks, using 7-day intervals.

We collected 2019 population data and administrative unit area data for each city from the 2019 Statistical Yearbook. Subsequently, we calculated the incidence rate and population density for each city. Population mobility data were obtained from Baidu Migration Data (https://qianxi.baidu.com/#/, accessed on 1 July 2020), which includes metrics such as intra-city travel intensity and a migration scale index for inflow and outflow. Given that population inflow is a significant influencing factor in the spread of the epidemic, we selected the inflow intensity index to represent population mobility data in our study.

### 2.2. Methodology

2.2.1. Heavy-Tailed Distributions

Heavy-tailed distributions [66–68] are characterized by a split between the mean of the data and the majority of the small values in the tail of the distribution, with only a minority of the large values in the head. The imbalance between the head and tail of heavy-tailed distributions can be conceptualized as the dominance of the small values over the large ones. They exhibit a feature of having a "fat head" and a "long tail", in statistical terms. A common visualization tool for exploring heavy-tailed distributions is the rank-size distribution plot (as shown in Figure 2), where ranks are plotted on the x-axis and corresponding values or sizes on the y-axis. Initially used in studies of word frequencies and city sizes [62], this visualization provides a straightforward and intuitive way to

investigate heavy-tailed distributions. Common forms of heavy-tailed distributions include power-law distributions, log-normal distributions, exponential distributions, and others.

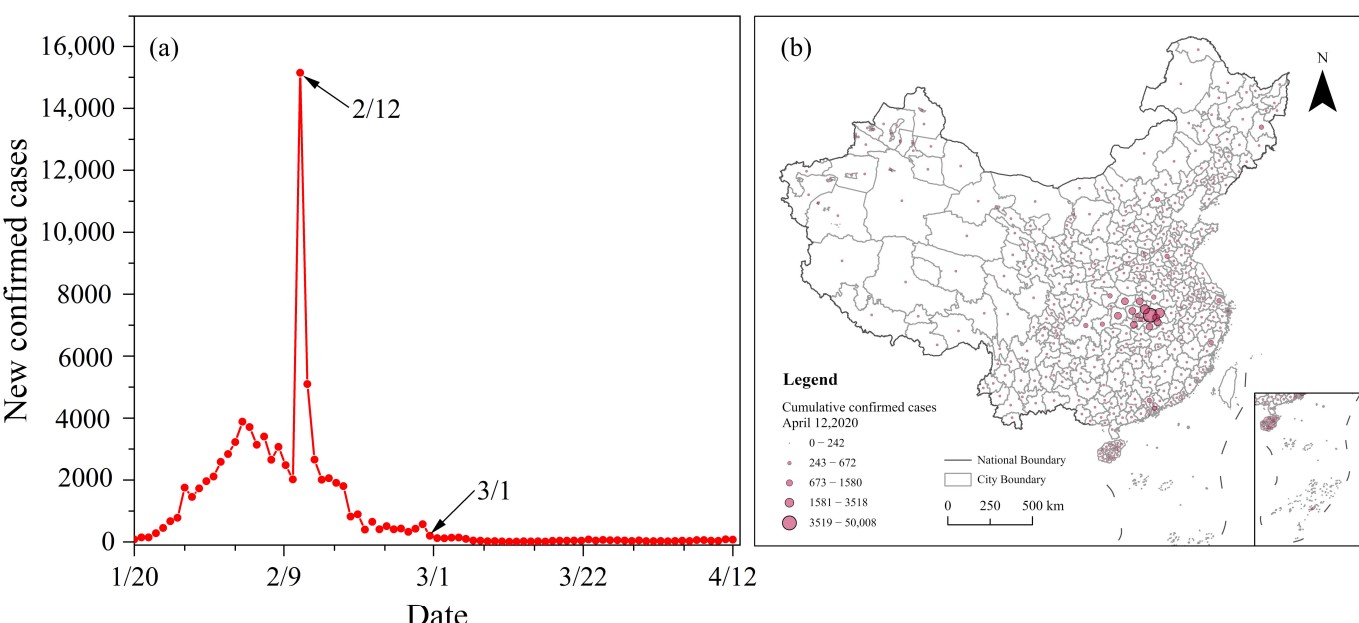

**Figure 1.** Development of the COVID-19 epidemic in China ((**a**) shows the time-series variation in new cases of COVID-19 in China; (**b**) shows the spatial distribution of the cumulative number of confirmed cases of COVID-19 as of 12 April 2020).

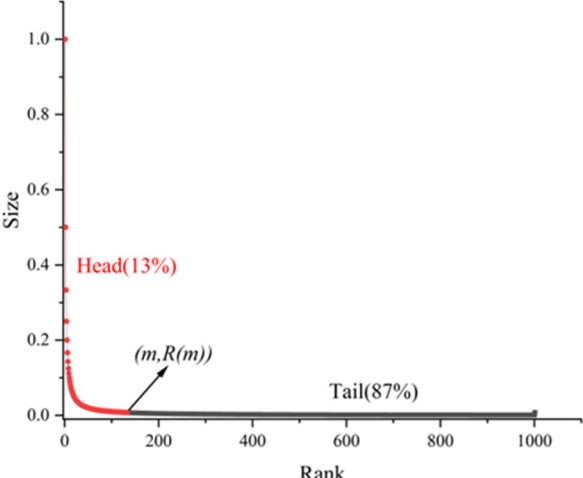

**Figure 2.** Rank-size plot (note: the x-axis is the sort and the y-axis is the size distribution of the corresponding sorted data values. The mean is m and the corresponding sort is R(m), with the head (e.g., 13%) and the tail (e.g., 87%)).

For a given dataset, specific statistical methods, such as power-law fitting [67], can be employed to detect whether it follows a heavy-tailed distribution. Power-law distributions reveal scaling relationships within data, indicating that small values occur much more frequently than large ones. The expression for a power-law distribution is given by:

$$y = cx^{-\alpha} \tag{1}$$

In this context, $\alpha$ represents the power-law exponent or scaling parameter ($1 < \alpha < 3$), and $c$ denotes a constant. Typically, a higher power-law exponent, $\alpha$, indicates greater

spatial heterogeneity. In other words, a power-law exponent of 3 implies more heterogeneity compared to an exponent of 1 [40].

As for power-law testing, the simplest approach involves taking the logarithm of both sides of Equation (1). The resulting function expression is as follows, and allows us to assess whether it forms a straight line on a double-logarithmic graph.

$$\ln y = -\alpha \ln x + \ln(c) \tag{2}$$

However, this method has limitations and relies on a substantial amount of data for reliability, being suitable primarily for discrete data. In this study, we employed a robust maximum likelihood approach and the Kolmogorov–Smirnov test to assess power-law distributions. The indicators of power-law fitting are represented by the estimated exponent $\hat{\alpha}$ and goodness-of-fit index $p$ A higher $p$ value indicates a better fit to the power-law distribution. A $p$ value approaching 1 suggests that the data closely follow a power-law distribution, with $p$ values exceeding 0.1 considered acceptable for fitting.

### 2.2.2. Head/Tail Breaks and ht Index

The head/tail breaks method was initially introduced by scholar Jiang Bin in 2013 as an approach to classify or stratify data with characteristics of heavy-tailed distributions [62], such as power-law distributions. It enables rapid data categorization and reveals the latent hierarchical structure of the dataset. The head/tail segmentation is a recursive function, as shown in Figure 3, used to iteratively deduce the head and tail for all levels (scales) and recursively unveil the inherent hierarchical structure of the dataset. Based on a descending order of data sorting, the original data are split into the head, consisting of values higher than the mean, and the tail, consisting of values lower than the mean. Since the tail exhibits minimal variation, there is no need to segment values below the mean. This process continues to iteratively split the head until the concept that small values are significantly more abundant than large values is violated, signifying that the head no longer conforms to a heavy-tailed distribution, at which point the segmentation stops.

The structural hierarchy levels obtained using the head/tail break method can be quantified using the ht index [69], revealing the number of levels in the hierarchy. The calculation formula involves adding 1 to the number of recursive segmentations. There are two fundamental principles governing hierarchical structures: scaling laws, where smaller levels are far more numerous than larger ones within the hierarchy, and Tobler's law, which stipulates the presence of varying degrees of similarity within each level of the rank structure. In other words, spatial heterogeneity and spatial homogeneity, or spatial dependency, can be simultaneously observed in the hierarchical structure. A higher ht index signifies a higher level of hierarchy within the structure.

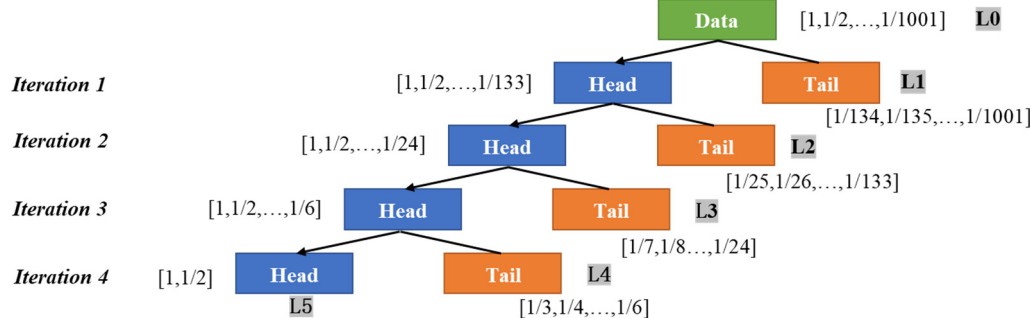

**Figure 3.** The recursive flow of head/tail break (the 1001 numbers [1, 1/2, ..., 1/1001] are divided into 5 levels, [1/134, 1/135, ..., 1/1001], [1/25, 1/26, ..., 1/133], [1/7, 1/8 ..., 1/24], [1/3, 1/4, ..., 1/6], [1, 1/2], with 5 inherent levels: L1, L2, L3, L4, L5).

We also employed the aforementioned set of 1001 data points (1, 1/2, 1/3, ..., 1/1001) to further illustrate this classification scheme (Figure 3). The average of the 1001 data points is 0.0075, which divides the 1001 data points into two categories: the head, consisting of data points greater than the average (1, 1/2, ..., 1/133), and the tail, consisting of data points lower than the average (1/134, 1/135, ..., 1/1001). For the data points in the head category (1, 1/2, ..., 1/133), their average is 0.0411. This value further divides the head data into two subcategories: the upper head (1, 1/2, ..., 1/24) and the lower head (1/25, 1/26, ..., 1/133). We continue this process with the upper head data, obtaining a third average value of 0.1573, which further divides the upper head data into two subcategories. Subsequently, we obtain a fourth average value of 0.4083, which further divides the six data points into two categories. After this, the data no longer follow a heavy-tailed distribution, and the segmentation process stops. In total, the data iterated (was segmented) four times, and the ht index is expressed as the number of recursive segmentations (as stated on the previous page) + 1, meaning the ht index is 5. The data are classified into five categories: [1/134, 1/135, ..., 1/1001], [1/25, 1/26, ..., 1/133], [1/7, 1/8, ..., 1/24], [1/3, 1/4, ..., 1/6], and [1, 1/2]. These correspond to five hierarchical structure levels: L1, L2, L3, L4, and L5. Furthermore, we can calculate the hierarchical level of each city in the total number of levels, i.e., $\frac{L_i}{N}$ = ht index rank, where $L_i$ represents hierarchical structure levels, and N is the ht index.

### 2.2.3. Epidemic Risk Measurement

This study provides a dynamic perspective for identifying the space-time risk of an epidemic in the space-time domain. We examine the disparity between the relative levels of cases and the population at each date for each specific region across the entire study area. The space-time risk refers to the likelihood of an outbreak around a specific region at a certain date. In this study, we define a risk assessment indicator, $R_{i,t}$, to characterize the space-time risk of COVID-19.

$$R_{i,t} = \frac{L_{i,t}^{(cas)}}{N_t^{(cas)}} - \frac{L_i^{(pop)}}{N^{(pop)}} \tag{3}$$

where $N_t^{(cas)}$ denotes the ht index of COVID-19 confirmed cases at the moment $t$, $L_{i,t}^{(cas)}$ denotes the hierarchical level of COVID-19 confirmed cases at the moment $t$, $N^{(pop)}$ is the ht index of the population density, $\frac{L_{i,t}^{(cas)}}{N_t^{(cas)}}$ denotes the $t$, which is the time rank of COVID-19 confirmed cases; $L_i^{(pop)}$ denotes the hierarchical level of the urban population density and $\frac{L_i^{(pop)}}{N^{(pop)}}$ represents the rank of the urban population. $R_{i,t}$ denotes the risk value for a certain region $i$ at time $t$. From Equation (3), we can obtain three possible outcomes: $R_{i,t} > 0$ indicates high risk, $R_{i,t} = 0$ indicates medium risk, and $R_{i,t} < 0$ indicates low risk. Obviously, if a region has a lower level of population density but a higher level of confirmed cases, the risk of epidemic spread exists. Specifically, while a region has a relatively low level of cases but a relatively high level of population density, $R_{i,t}$ is negative, which indicates low risk. While the two relative levels of cases and population density are identical, $R_{i,t}$ equals zero and indicates medium risk. Further, while the relative level of the population density is low but the relative level of cases is high, $R_{i,t}$ is positive, which indicates high risk. Thus, if the risk assessment indicator, $R_{i,t}$, is greater than 0, it indicates the epidemic risk of a certain city $i$ at a certain date $t$ (a higher value of $R_{i,t}$ indicates increased risk). It is worth noting that this study does not consider the situations of medium and low risk, focusing on the spatiotemporal variations in high-risk areas. The positive value of the risk assessment indicator represents the epidemic risk across the space-time domain. Its greater value indicates a higher risk. Specifically, when a region exhibits the highest hierarchical level of cases at a certain date, and meanwhile exhibits the lowest level of population density, the epidemic risk reaches the highest

level (approaching 1). According to Equation (3), the highest risk value in this study was calculated as up to 0.84.

## 3. Results

### 3.1. Exploration of Heavy-Tailed Distribution

We first examined the heavy-tailed distributions of COVID-19 cases and population density. The results indicated that both exhibited heavy-tailed distributions. According to statistics, the total population density of 367 Chinese cities in 2019 was 148,161.80 individuals per square kilometre, with an average population density of 403.71 individuals per square kilometre. Among these, the highest population density was 6727.91 individuals per square kilometre, while the lowest was 0.35 individuals per square kilometre. Table 1 displays the results of population density head/tail break calculations, involving five iterations of the population density data. Figure 4a,b depict the top three stratification levels of the population density and the magnitude ranking of the confirmed COVID-19 cases in the first week, respectively, using embedded insets. The head data are represented in red, while the tail data are in blue. Two smaller insets are added for each dataset, providing a visual representation of the heavy-tailed distribution characteristics of the data and further illustrating the role of the data hierarchy structure. As shown in Figure 4a, the largest plot encompasses population density data for all 367 cities. Dark red indicates the first head (126 cities), while dark blue represents the first tail (241 cities). Replotting the population density rank-size plot for the 126 cities in the first head reveals that dark red denotes 30 cities with a population density in the head, while dark blue represents 96 cities with a population density in the tail. The smallest plot is for the population density rank-size plot within the second head, consisting of 30 cities, and it also exhibits heavy-tailed characteristics in the population density data.

Throughout the power-law fitting process, except for the first week, the power-law exponent $\alpha$ remained around 2, with goodness-of-fit ($p$) values consistently exceeding 0.1. We did a power-law estimation test for the number of confirmed COVID-19 cases at 12 weeks (Figure 4c–e), and as shown in Figure 4d,e, the maximum value of the power-law estimation index for COVID-19 cases ($\alpha = 2.02$) occurred in the first week, while the minimum value of the goodness-of-fit p ($p = 0.048$ ($p < 0.05$)) also occurred in this time period, and it did not pass the power-law test. The COVID-19 confirmed cases did not show a power-law distribution in the first week. However, the pattern of power-law distribution was presented after only the second week, and the power-law distribution of the COVID-19 confirmed cases showed a trend of gradually decreasing significance with increasing time (Figure 4c: change from dark blue to light blue).

In the context of the dynamic development of the epidemic, the power-law estimation index $\alpha$ underwent an overall decreasing and then increasing evolution, with an inverted "V" shape (Figure 4d). The most robust spatial heterogeneity was observed for COVID-19 cases in the second week ($\alpha = 2.02$, $p > 0.05$) and the smallest in the seventh week ($\alpha = 1.58$, $p > 0.05$). We can find that, as the power law estimation index decreases, the scale effect of the epidemic spread decreases and so does the spatial heterogeneity. At the beginning of the epidemic, the local outbreak was at the "low level of spread" stage, the scale effect of the epidemic spread was not obvious, and there was no power-law distribution. With the spread of the epidemic, the COVID-19 cases gradually showed a power-law distribution. It is worth clarifying that the confirmed cases of COVID-19 did not show a power-law distribution pattern in the first week, but it is clear from the result of their head/tail breaks (Table 1) and rank-size plot (Figure 4b) that they also satisfy the heavy-tailed distribution characteristics.

**Table 1.** Population density and calculation of head/tail breaks of confirmed COVID-19 cases in week 1.

|  | City | Mean | Head | Tail | %Head |
|---|---|---|---|---|---|
| Population density | 367 | 403.71 | 126 | 241 | 34% |
|  | 126 | 861.24 | 30 | 96 | 24% |
|  | 30 | 1668.05 | 8 | 22 | 27% |
|  | 8 | 3150.71 | 3 | 5 | 38% |
|  | 3 | 4666.17 | 1 | 2 | 33% |
| First week COVID-19 confirmed case | 367 | 6.96 | 70 | 297 | 19% |
|  | 70 | 29.99 | 17 | 53 | 24% |
|  | 17 | 87.06 | 5 | 12 | 29% |
|  | 5 | 188.4 | 1 | 4 | 20% |

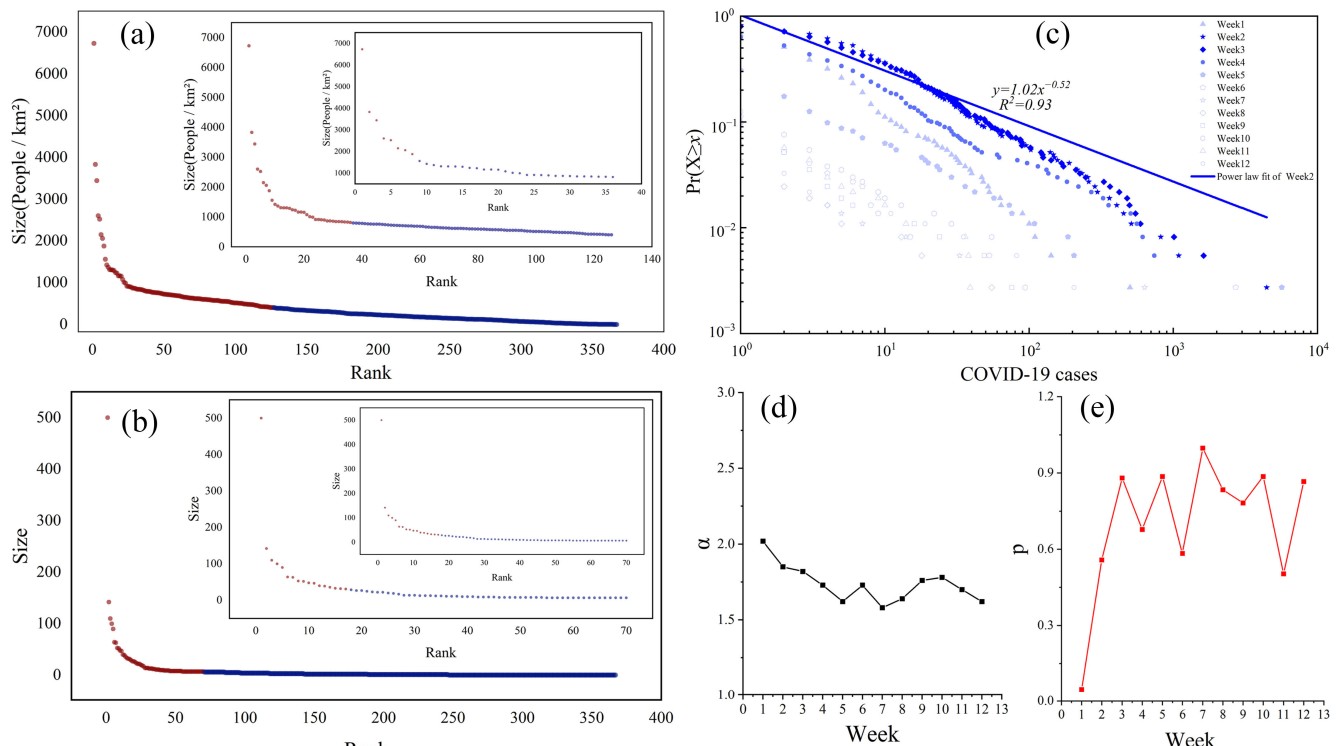

**Figure 4.** Population density and the heavy-tailed distribution of COVID-19 confirmed cases ((**a**,**b**) show the first three levels of rank-size plots for population density and first-week COVID-19 confirmed case head/tail break results, respectively; (**c**) shows the power-law variation for 12 weeks of COVID-19 confirmed cases; (**d**,**e**) show the trend of power-law index α and goodness-of-fit p, respectively).

## 3.2. The ht Index and Spatial Hierarchy of COVID-19 Confirmed Cases and Population Density

We can use head/tail breaks to derive the hierarchical structures for population densities and the numbers of confirmed COVID-19 cases (See Appendix A for full details). The ht index indicates the spatial hierarchy structure. For example, the population density ht index = 6, implying the presence of six hierarchical levels, namely L1 (0–404), L2 (405–861), L3 (862–1668), L4 (1669–3151), L5 (3152–4666), and L6 (4667–6728). From Figure 5, it can be found that, for the population densities, the hierarchical structure had a stable ht index value of 6. Or, we can say that the spatial hierarchy of the population densities had a total of six levels. We employed choropleth symbolization to visualize the spatial hierarchy levels of the population density for each city. The darker the color, the higher the spatial hierarchy level, indicating that L1 (pop: 0–404) < L2 (pop: 405–861) < L3 (pop: 862–1668) < L4 (pop: 1669–3151) < L5 (pop: 3152–4666) < L6 (pop: 4667–6728). As depicted in Figure 6, cities with a higher population density are generally located in the eastern regions, particularly in some coastal areas such as Shenzhen

and Shanghai. For the epidemic cases (Figure 5), the ht index value varied from 3 to 5. The first week had the largest ht index value of 5. At first, there were a total of five levels in the spatial hierarchy of the cases, and then, the ht index varied around 3 and 4 during the entire period.

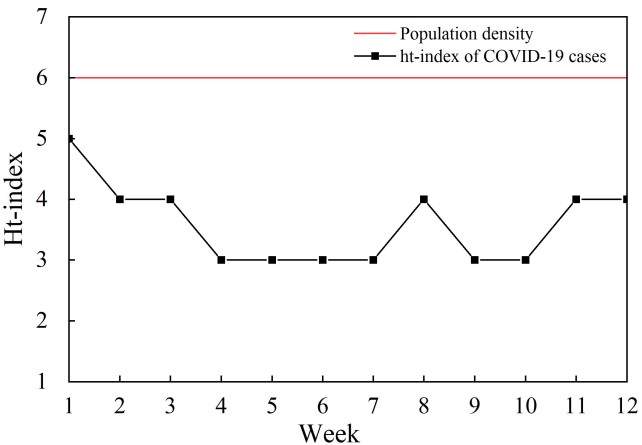

**Figure 5.** Time-series variation in population density and ht index of COVID-19 confirmed cases.

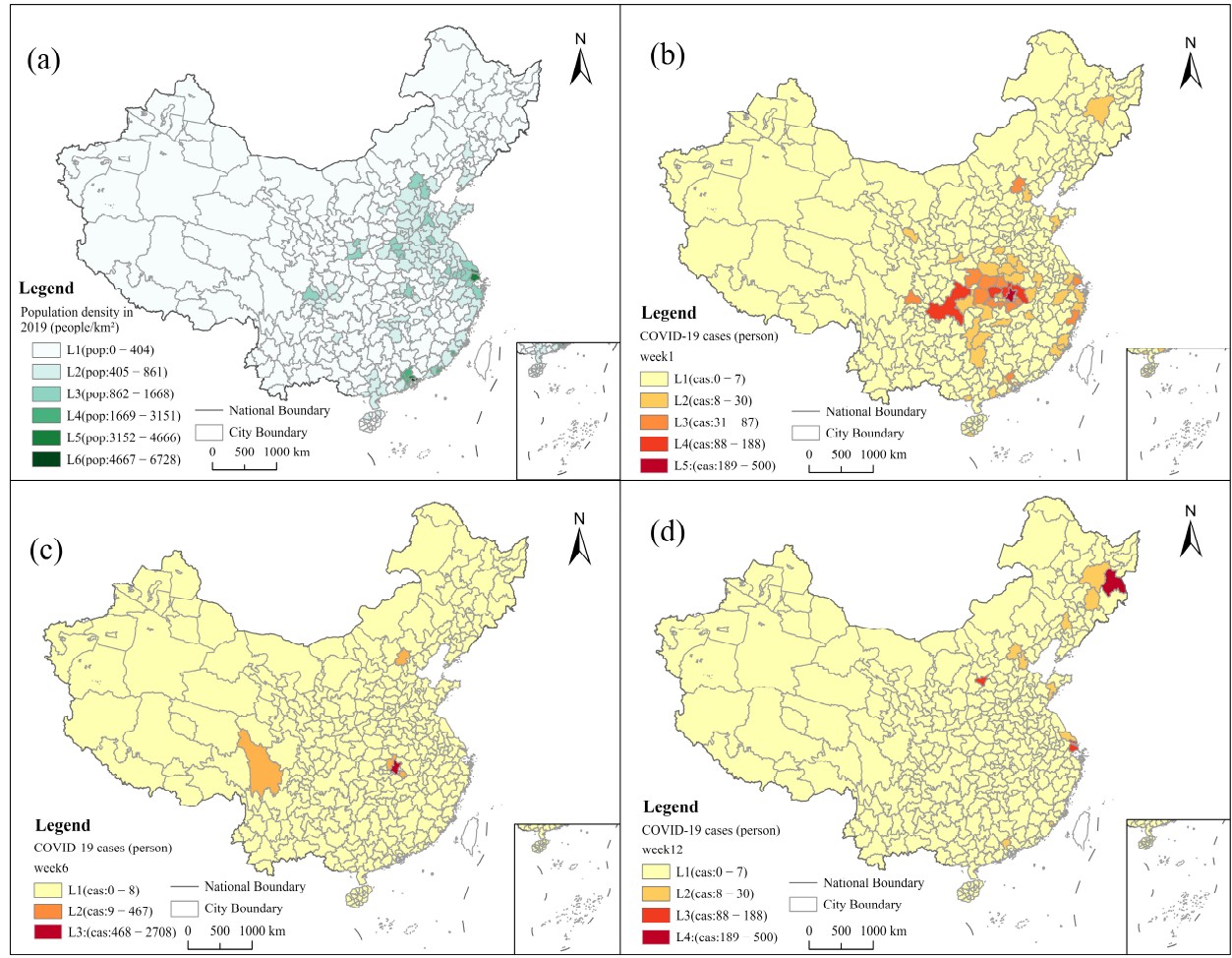

**Figure 6.** Geographical distributions of the spatial hierarchy of population density and COVID-19 confirmed cases ((**a**) shows the hierarchical structure of the spatial hierarchy of population density; (**b**–**d**) show the change in the hierarchical structure of the spatial hierarchy in the 1st, 6th, and 12th weeks, respectively).

To explore the inherent spatial hierarchy of COVID-19 cases, we further generated spatial hierarchy maps for the first week (ht index = 5), the sixth week (ht index = 3), and the 12th week (ht index = 4) in relation to the number of COVID-19 cases (Figure 6b–d). Darker colors indicate higher spatial hierarchy levels of the cities. In Figure 6b, during the first week, the cities with higher levels of COVID-19 cases were concentrated around Wuhan, with five cities being at the highest level, namely Wuhan (500 cases), Huanggang (142 cases), Chongqing (110 cases), Xiaogan (100 cases), and Jingmen (90 cases). By the sixth week (Figure 6c), the cities with higher COVID-19 case numbers remained concentrated around Wuhan, but the hierarchy structure of the case numbers decreased to three levels, indicating a reduction in the heterogeneity of the pandemic spread. After nearly three months of epidemic prevention and control efforts, in the final week of the study (Figure 6d), the ht index for the spatial hierarchy structure of the cases was 4. Overall, the COVID-19 case levels in the eastern region were higher than those in the western region, and the hierarchy structure changed over time. For instance, from the first week to the sixth week, the levels decreased from Wuhan, then in surrounding cities, and then in other areas. By the 12th week, the cities with higher levels were distributed in the northeast region.

### 3.3. Assessment of the Risk of Transmission of COVID-19 Cases in China

Based on the hierarchical structures of the population density and COVID cases, we can calculate the risk assessment indicators for all cities over 12 weeks and display the spatiotemporal dynamics of high-risk cities. Figure 7 illustrates the spatial distribution of high-risk cities from week 1 to week 12. The color red represents risk, with darker shades indicating a higher risk. The evolution of the COVID-19 space-time risk exhibits significant regional disparities. During the early stages of the study, i.e., from week 1 to week 5 (Figure 7a–d), the high-risk cities were predominantly concentrated around Wuhan, displaying a larger "single-core" structure. However, in the middle weeks, from week 6 to week 8 (Figure 7f–h), the distribution of the high-risk cities became more scattered, with only a few cities exhibiting relatively higher infection risks. Increases in population mobility may contribute to higher epidemic risks. In the final four weeks, i.e., from week 9 to week 12 (Figure 7i–k), the high-risk cities were primarily located in the eastern coastal regions and a few central-western areas, such as Beijing and Chongqing. These cities exhibit high population density, intense population migration, and vibrant socioeconomic activity, resulting in elevated risk values.

To delineate more detailed information about the pandemic risk, we primarily investigate the changes in the number of cities at risk from week 1 to week 12, along with the statistical metrics of the average, standard deviation, and maximum risk values per week. The average is the sum of the space-time risks of all cities divided by the number of cities in each week, which reflects the trend of the space-time risk concentration. Standard deviation mathematically refers to the arithmetic square root of the arithmetic mean of the squared deviation from the mean (i.e., variance). In this study, computing the standard deviation of the risk values over the 12 weeks serves as a measure of uncertainty, representing the precision of the risk estimates. The maximum risk value per week is the one with the largest weekly temporal risk for all cities. Over the entire study period, the number of high-risk cities decreased from 57 in the first week to 6 in the last week (Figure 8). However, it is worth noting that, in the last four weeks, the average value and standard deviation of the risk assessment indicator noticeably increased. While the number of high-risk cities decreased significantly during the study period, greater attention should be paid to the later stages of epidemic control, especially in several cities with extremely high-risk values. It is noteworthy that the maximum risk value occurred in the last three weeks.

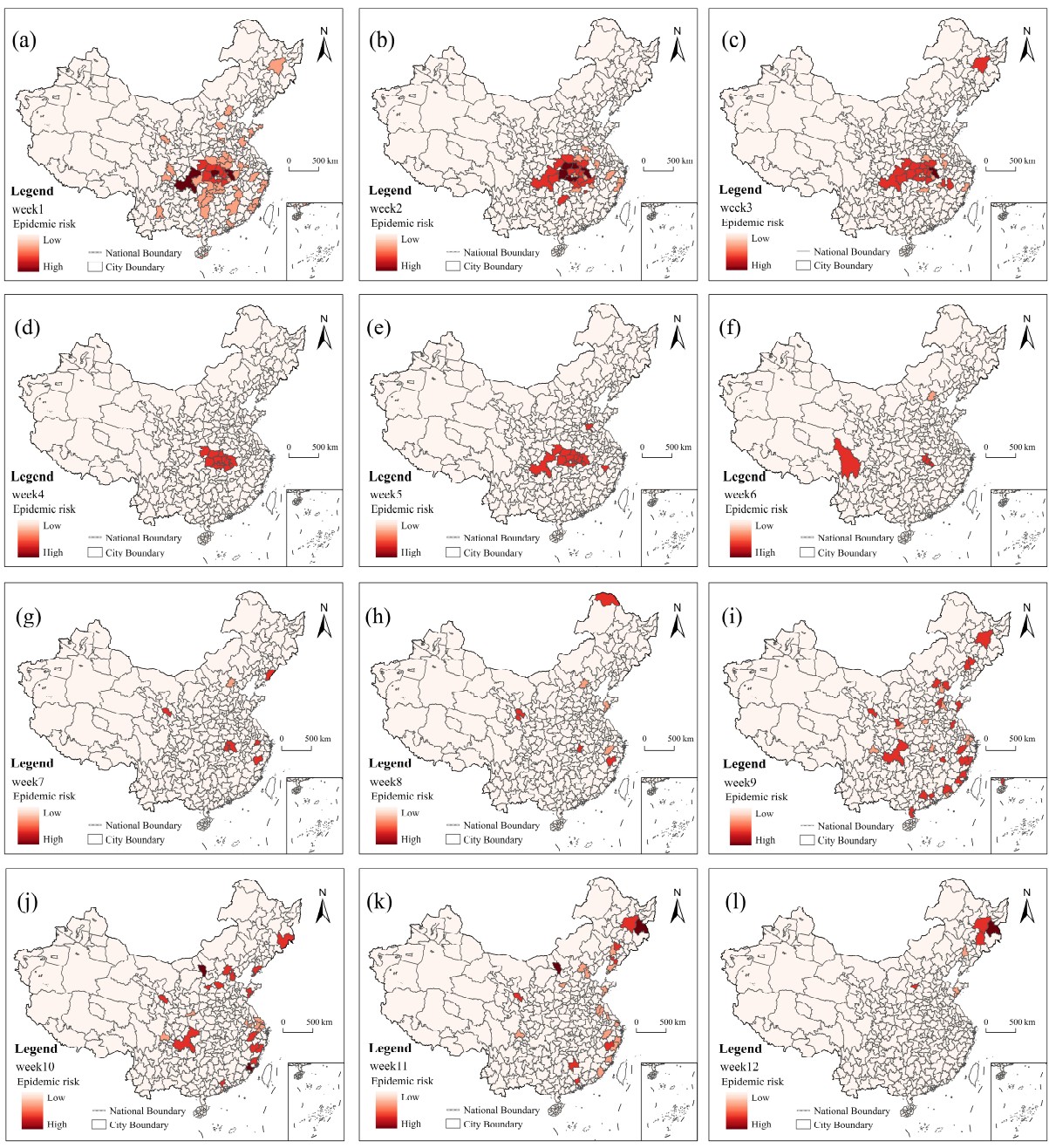

**Figure 7.** Geographical distributions of high-risk cities (note: (**a**–**l**) represent the spatial distributions of high-risk cities from week 1 to week 12, respectively; the darker the colour, the higher the risk).

### 3.4. Application of Outbreak Risk Assessment Models

Finally, we selected six representative cities: Beijing, Guangzhou, Wuhan, Shanghai, Wenzhou, and Chongqing, to explore their local-level risk dynamics and compared their 84-day risk assessment values with their incidence rates and population migration intensities (Figure 9). We observed that the risk evolution in these six cities varies, and their relationship with the incidence rate and population migration intensity differs as well.

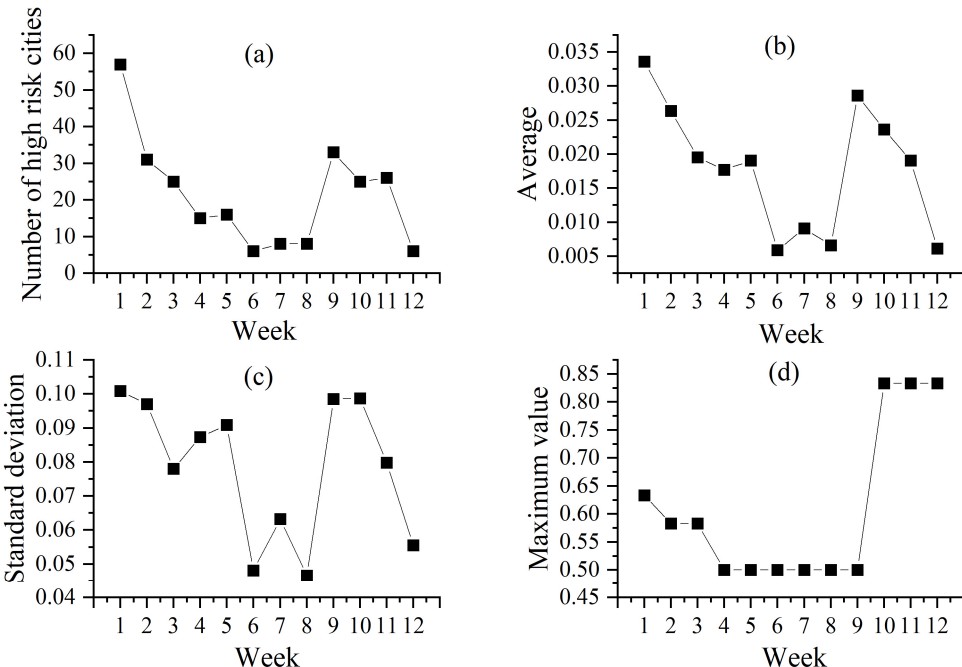

**Figure 8.** Numbers of high-risk cities and statistics of the risk assessment indicator during 12 weeks (note: (**a**–**d**) show the number of cities at risk, mean, standard deviation, and maximum risk value changes per week, respectively).

An increase in the incidence rate contributes to a higher infection risk in a region, which is particularly evident in the early stages of the epidemic (the rapid growth phase and the control reduction phase), as seen in Wenzhou from 26 January to 11 February (Figure 9e) and Chongqing from 21 January to 26 January (Figure 9f). In the middle and later stages of the epidemic, a city's epidemic risk is mainly influenced by its population mobility. When the incidence rate is low, a city may still face a high risk due to factors such as population mobility. Beijing (Figure 9a) showed a higher population migration rank from 28 March to 10 April, increasing the local epidemic risk as the influx of people grew. From 18 March to 9 April, Wenzhou experienced a gradual rise in its population inflow intensity, reaching a high of 0.42 during that period. Chongqing exhibited similar characteristics from 20 February to 23 February. Furthermore, due to the virus's high destructiveness, high infection rate, and rapid transmission, the risk persists even with the implementation of government-enforced lockdown measures. This is particularly evident in the Wuhan region (Figure 9c), where the risk remained consistently high, especially from 20 January to 17 March.

It is worth noting that, due to the lag in epidemic risk assessment, situations can occur where risks are not immediately recognized during periods of high incidence rates and high population migration ranks. Instead, risks may manifest later, especially as observed in Guangzhou from 26 January to 6 February (Figure 9b) and Shanghai from 25 January to 4 February (Figure 9d). Therefore, in epidemic control efforts, vigilance should not be relaxed even when the incidence rate is high and the population influx is significant, to prevent potential epidemic rebounds.

In summary, the risk dynamics in these six cities vary, and their relationships with incidence rates and population migration intensity differ as well. The approach we have presented allows for a better description of the risk dynamics in each city. Specifically, an increase in the incidence rate can elevate the risk for a city. However, it is crucial to recognize that, even with lower incidence rates, other factors such as population mobility can still place a city at a high risk.

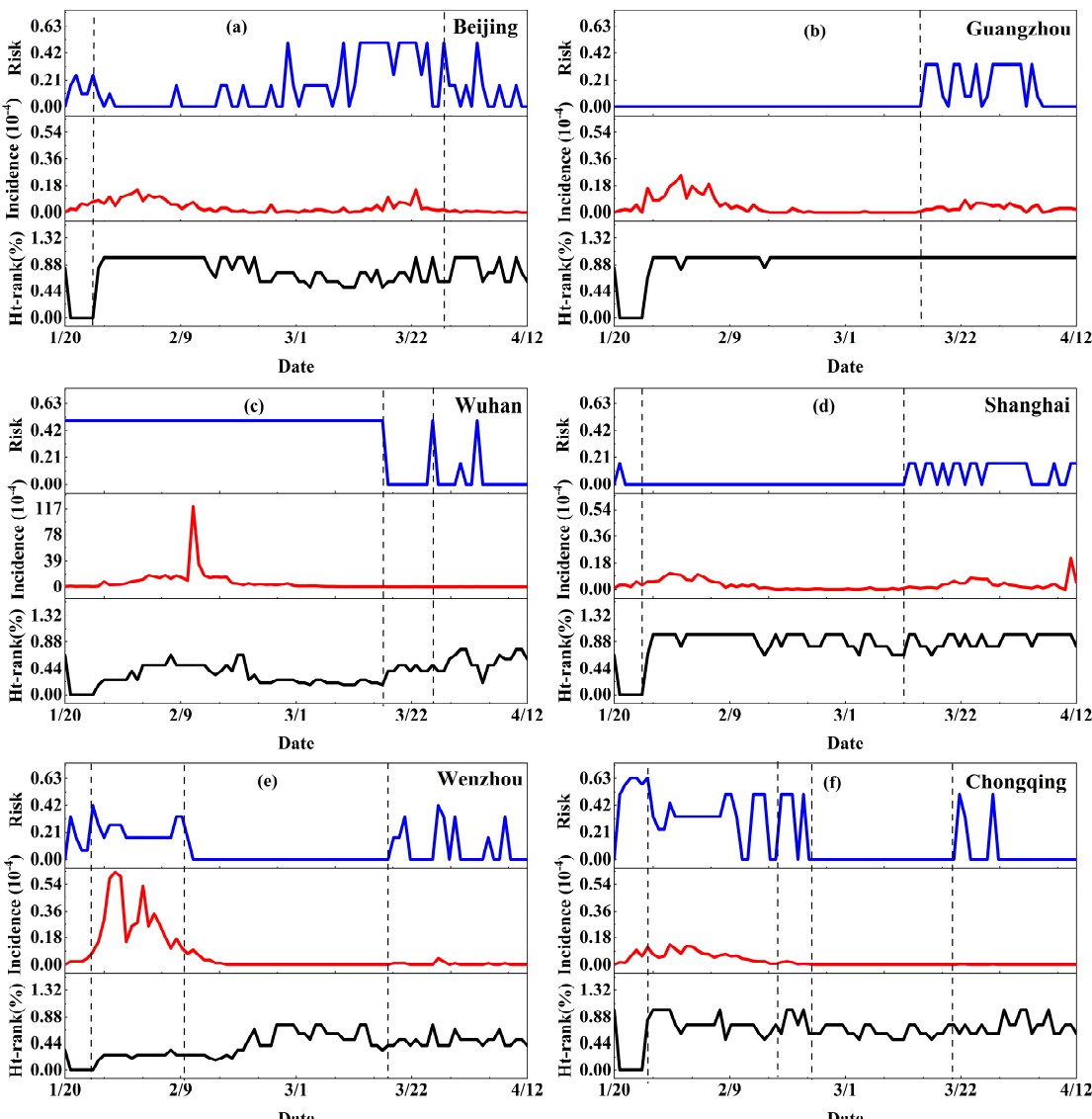

**Figure 9.** Time-series variation in risk, incidence, and the ht index rank of population migration intensity in six cities ((**a**–**f**) represent: Beijing, Guangzhou, Wuhan, Shanghai, Wenzhou, Chongqing; blue indicates risk values $R_{i,t}$, red indicates prevalence, black indicates population migration intensity ht index rank).

## 4. Discussion

Epidemic risk assessment has been a focal point of research in post-pandemic China and other countries. In recent years, the head/tail break method has been widely applied across various domains, including urban studies [62,64], environmental pollution studies [70], and more, offering functions such as image texture extraction and data visualization. In epidemiology, the head/tail break method is commonly employed as a classification approach to visualize the spread patterns of diseases, such as the mapping of the global COVID-19 pandemic [40].

This study employed the head/tail segmentation method to investigate the space-time evolution of COVID-19 risks in China. The COVID-19 pandemic is influenced by underlying populations, and the spatial hierarchies of epidemic cases in different regions correlate with the spatial hierarchy of the base population. By comparing the differences in the relative hierarchy of the population density for confirmed COVID-19 cases, a risk measurement model was constructed. This proposed risk measurement model quantifies the impact of population density on infectious disease transmission and is feasible for

identifying areas with a high risk of infection in urban epidemic surveillance. Important research results were achieved, providing valuable references for cities facing increasing cases during the pandemic.

In the early stages of the new coronary pneumonia outbreak, many researchers have predicted the potential outbreak size and duration through various models [71–73]. For instance, the basic reproduction number ($R_0$), serving as a crucial indicator of infectious disease transmission, elucidates the extent of COVID-19 case dissemination. In real-time studies of epidemic transmission risk, the effective reproduction number (Rt) is employed to signify the current epidemic transmission index, which studies have used to provide some useful clues for health authorities in China and other countries or regions. In contrast, one of the significant contributions of our study is the introduction of a method to directly depict epidemic risks. Based on two parameters, namely newly confirmed COVID-19 cases and population density, we constructed a risk measurement model that dynamically assesses the risk of epidemic spread through spatially stratified ranking. Compared with other epidemic risk assessment models such as SEIR models and Kalman filtering, the risk assessment indicator proposed in this paper, $R_{i,t}$, is more straightforward and user-friendly.

Our study of one-week units reveals the risk of COVID-19 transmission in China. It is worth stating that the virus does not adhere to or respect either aggregation definition. Hence, different time periods (e.g., 10-day or 14-day) would likely produce different results. Moreover, a different spatial structure (e.g., at a coarser administrative level) would also produce different results. In other words, the results produced by this article are only valid under the assumption of a 7-day week and the municipal administrative district. Other temporal frameworks or enumeration unit boundary structures for the same period (12 weeks) and geographic area (China) will result in different outcomes. Thus, the effects of the modifiable areal unit problem (MAUP) cannot be ignored. We observed that areas with higher pre-epidemic levels may have a higher risk of infection. The number of high-risk cities increased sharply, mainly concentrated in densely populated coastal cities, central and western regions, and the Beijing-Tianjin-Hebei region. This is because these areas have high population densities and active socio-economic activities. Therefore, the timely identification of densely populated areas should be considered one of the key measures in controlling the spread of the disease [74,75]. After the outbreak, various regions implemented a series of targeted infection prevention and control measures, including isolating confirmed cases and suspected cases, breaking the transmission chain, and reducing the spread of COVID-19 in various regions. As a result, the epidemic in China was well-controlled within a short period (6–8 weeks).

In addition, the results of the risk assessment not only reflect the inherent hierarchy of the epidemiological and demographic context at the macro level but also portray the dynamics of risk evolution at the micro level for local areas or individual cities. The analysis of the daily risk assessment values of six cities, Beijing, Guangzhou, Wuhan, Shanghai, Wenzhou, and Chongqing, compared with their COVID-19 incidence rates and the ranking of their population movement intensity, reveals that the model has a certain degree of foresight and can effectively assess the risk of an area in the future. When the incidence rate rises, the risk value will also rise. The risk of an epidemic is influenced by population movement, and an increase in the rank of the population inflow will bring about the risk of an epidemic, and the risk value will fluctuate and change. To a certain extent, this provides the basis for the government to adjust and update its strategy and shorten the time for China to prevent and control the epidemic, which provides important support for the rapid resumption of production in February and March.

Another scientific contribution of this study lies in the pioneering application of the head/tail breaks approach to the space-time risk research of epidemics. The proposed method offers a better depiction of the risk evolution in each city. The study visually mapped the space-time risk evolution of each city over 12 weeks. In terms of spatial distribution, the high-risk cities were concentrated around Wuhan in the early stages, forming a "large core," and were later distributed in coastal areas, forming a "striped"

pattern. To validate the relationship between the space-time risk changes in each city and the actual situation, we compared the incidence rates, the ht index ranking of the population influx intensity, and the $R_{i,t}$ values of six cities. We found that the risk assessment index has a certain usability and can be applied to the evaluation of urban epidemic risk. This model has provided a new perspective on the assessment of epidemic transmission risk levels and a new approach to the analysis of space-time changes in epidemic risk.

Although we have revealed spatial and temporal variation in areas at risk of outbreaks in China, further research is needed. As many stringent interventions have been implemented in some cities, more potential variables (preventive and control measures, vaccination, virus transmission characteristics, etc.) should be considered to refine the identification of areas at a high risk of transmission of confirmed COVID-19 cases. Secondly, longer-period datasets of the COVID-19 outbreak in China could be incorporated to further enhance the depth and significance of our research, and time-series data of the COVID-19 outbreaks in other countries or regions should be collected to validate the study in this paper. Exploring data on the impact of multiple factors on the spread of the epidemic and long-time series epidemics is an important research component to improve the accuracy of risk assessment and prediction.

## 5. Conclusions

This study, taking China as an example, analyzes the spatiotemporal characteristics of the evolution of COVID-19 risk over 12 weeks from 20 January to 12 April 2020, in 367 prefecture-level cities, focusing on the perspective of spatial hierarchy differences. We propose a COVID-19 risk assessment index based on the head/tail break method to identify the spatiotemporal distribution of high-risk cities. In this research, we observed that both COVID-19 cases and the population density exhibit heavy-tailed distribution characteristics. We found that the hierarchy structure of COVID-19 cases in China consistently approached but never surpassed the hierarchy structure of the population density. Consequently, the spatiotemporal transmission of COVID-19 is influenced by the underlying population. Furthermore, the calculated risk assessment values have practical significance, as they increase with rising incidence rates. It is worth noting that, even with lower incidence rates, the influence of other potential factors like population mobility can place an area at high risk.

In summary, these findings enhance the overall understanding of the space-time risk of COVID-19 in China and provide a scientific basis for policymakers. The risk measurement model we propose can describe the spatiotemporal patterns of COVID-19 risk and offers a new perspective for assessing the risk level of epidemic transmission, which holds practical significance. However, this study did not delve into more detailed influencing factors of epidemic risk, and the time series is relatively short. Therefore, future research should focus on longer time series and a comprehensive consideration of factors influencing changes in epidemic risk.

**Author Contributions:** Conceptualization, Bisong Hu and Shuhua Qi; methodology, Bisong Hu; software, Tingting Wu and Bisong Hu; validation, Bisong Hu and Jin Luo; formal analysis, Tingting Wu, Jin Luo and Bisong Hu; investigation, Shuhua Qi and Jin Luo; resources, Shuhua Qi and Bisong Hu; data curation, Tingting Wu, Bisong Hu and Jin Luo; writing—original draft preparation, Tingting Wu; writing—review and editing, Tingting Wu and Bisong Hu; visualization, Tingting Wu, Jin Luo and Shuhua Qi; supervision, Bisong Hu; funding acquisition, Bisong Hu. All authors have read and agreed to the published version of the manuscript.

**Funding:** This research was funded by the National Natural Science Foundation of China (grant number 42061075) and the Graduate Innovation Fund of Jiangxi Normal University, China (No. YJS2022013). The funders have no role in study design, data collection and analysis, decision to publish, or preparation of the manuscript.

**Data Availability Statement:** The spatiotemporal data of daily new confirmed cases were collected from multiple official and publicly available sources, and had been comparatively verified the epidemic data through the public platform of the 2019-nCoV-infected pneumonia epidemic (http://2019ncov.chinacdc.cn/2019-nCoV/). The dataset of China's urban populations is owned by China Population Census Yearbook 2020 (http://www.stats.gov.cn/tjsj/pcsj/rkpc/7rp/zk/indexce.htm). The head/tail breaks software can be found at https://en.wikipedia.org/wiki/head/tail_breaks.

**Conflicts of Interest:** The authors declare no conflict of interest. The funders had no role in the design of the study; in the collection, analyses, or interpretation of data; in the writing of the manuscript, or in the decision to publish the results.

## Appendix A

This appendix includes a statistical table of the results of head/tail breaks calculations for 12-week COVID-19 cases.

**Table A1.** Results of head/tail break COVID-19 cases calculations.

| Date | City | Mean | Head | Tail | %Head |
|---|---|---|---|---|---|
| Week1 | 367 | 6.96 | 70 | 297 | 19% |
| | 70 | 29.99 | 17 | 53 | 24% |
| | 17 | 87.06 | 5 | 12 | 29% |
| | 5 | 188.40 | 1 | 4 | 20% |
| Week2 | 367 | 39.31 | 38 | 329 | 10% |
| | 38 | 312.71 | 7 | 31 | 18% |
| | 7 | 1155 | 1 | 6 | 14% |
| Week3 | 367 | 62.88 | 30 | 337 | 8% |
| | 30 | 669.23 | 3 | 27 | 10% |
| | 3 | 4798 | 1 | 2 | 33% |
| Week4 | 367 | 85.75 | 15 | 352 | 4% |
| | 15 | 1971.33 | 1 | 14 | 6% |
| Week5 | 367 | 18.73 | 16 | 351 | 4% |
| | 16 | 412.63 | 1 | 15 | 6% |
| Week6 | 367 | 7.84 | 6 | 361 | 2% |
| | 6 | 467 | 1 | 5 | 17% |
| Week7 | 367 | 1.93 | 9 | 358 | 2% |
| | 9 | 78 | 1 | 8 | 11% |
| Week8 | 367 | 0.31 | 15 | 352 | 4% |
| | 15 | 7.47 | 3 | 12 | 20% |
| | 3 | 28.67 | 1 | 2 | 33% |
| Week9 | 367 | 0.63 | 36 | 331 | 10% |
| | 36 | 6.42 | 4 | 32 | 11% |
| Week10 | 367 | 1.02 | 28 | 339 | 8% |
| | 28 | 12.68 | 6 | 22 | 21% |
| Week11 | 367 | 0.59 | 37 | 330 | 10% |
| | 37 | 5.84 | 8 | 29 | 22% |
| | 8 | 20.25 | 3 | 5 | 38% |
| Week12 | 367 | 1.04 | 12 | 355 | 3% |
| | 12 | 30.83 | 3 | 9 | 25% |
| | 3 | 107.33 | 1 | 2 | 33% |

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
