# Peer review of "A Head/Tail Breaks-Based Approach to Characterizing Space-Time Risks of COVID-19 Epidemic in China’s Cities"

_ijgi, doi:10.3390/ijgi12120485_

Round 1

Reviewer 1 Report

Comments and Suggestions for Authors

The work was done according to the instructions given in the instructions for the magazine. The organization of the work by sections (1. Introduction, 2. Data and methodology, 3. Results, 4. Discussion, 5. Conclusion) is according to the recommendations (Introduction, Materials and methods, Results, Discussions, Conclusions) and adequate with. The complete material is ordered in a way that is logical, clear and easy to follow.

The authors have adequately and appropriately cited sources, and all citations in the text are listed in the References section. English language and style are fine.

The work presents a new approach to the characterization of spatio-temporal risks from an epidemic.

The data analysis was done excellently, the results and research data from the field of research and experiments are presented and visualized in a clear way, which makes the work readable. The discussion is very detailed. My only objection here is that there is no comparison with other approaches in the literature.

There are no new methods or techniques, but it is a very well studied description, which gives a new perspective. It is not clear what scientific contribution the author adds to this article.

I would recommend the authors to describe in the article what is the scientific contribution of their work.

Author Response

Point 1: The work was done according to the instructions given in the instructions for the magazine. The organization of the work by sections (1. Introduction, 2. Data and methodology, 3. Results, 4. Discussion, 5. Conclusion) is according to the recommendations (Introduction, Materials and methods, Results, Discussions, Conclusions) and adequate with. The complete material is ordered in a way that is logical, clear, and easy to follow. The authors have adequately and appropriately cited sources, and all citations in the text are listed in the References section. English language and style are fine.

Response 1: We sincerely thank you for the appreciation of our work. We have done our best to improve the quality of the manuscript according to the reviewers’ comments. The one-to-one responses are laid out as follows. All the modifications are highlighted in blue in the revised manuscript. Hope the revision can address your concern.

Point 2: The work presents a new approach to the characterization of spatio-temporal risks from an epidemic. The data analysis was done excellently, the results and research data from the field of research and experiments are presented and visualized in a clear way, which makes the work readable. The discussion is very detailed. My only objection here is that there is no comparison with other approaches in the literature.

Response 2: Thanks for your appreciation again. This article aims to provide a novel perspective for identifying the risk of an epidemic in the space-time domain, which is different from the perspective in epidemiology. This revised manuscript has included the discussion about our approach compared with other epidemiological indicators such as basic reproduction number (Page 15, Lines 433-437, 439-444).

Point 3: There are no new methods or techniques, but it is a very well studied description, which gives a new perspective. It is not clear what scientific contribution the author adds to this article. I would recommend the authors to describe in the article what is the scientific contribution of their work.

Response 3: We appreciate your constructive comments. In this revised manuscript, we have included the discussion about the contributions of this article (e.g., Page 15, Lines 444-451; Lines 484-495). Hope the revision can address your concern.

Reviewer 2 Report

Comments and Suggestions for Authors

This work employed the data of confirmed cases in the initial stage of the COVID-19 outbreaks and furnished a head/tail breaks-based method to qualify the space-time risks of COVID-19 in cities of China. Several aspects may/should be addressed to make the article more convincing.

1. The discussion of equation (3) which is the main contribution of the paper seems to be deficient. What is the feasible range of R_i,t? How high a R_i,t value indicates a high space-time risk? Is it possible to have a negative or zero R_i,t value, if so, how to express the negative R_i,t value or 0 value?

2. Authors may need to give a clear description or definition of the “space-time risk”, otherwise, readers may possibly not understand this risk index well. For example, the basic reproductive number R_0 has very clear meanings.

3. The sample data only covers the very initial stage of COVID-19 outbreaks, during which rigorous quarantine measures were taken in China mainland. If longer period datasets can be included, the work will be wonderful.

4. The 12 pictures in figure 8 indicate significant risk variances in space and time. Are these results consistent with the reality? It may be better to include such discussions.

Comments on the Quality of English Language

Minor editing of English language required

Author Response

We sincerely thank you for the appreciation of our work. We have done our best to improve the quality of the manuscript according to the reviewers’ comments. The one-to-one responses are laid out as follows. All the modifications are highlighted in blue in the revised manuscript.

Point 1: This work employed the data of confirmed cases in the initial stage of the COVID-19 outbreaks and furnished a head/tail breaks-based method to qualify the space-time risks of COVID-19 in cities of China. Several aspects may/should be addressed to make the article more convincing. 1. The discussion of equation (3) which is the main contribution of the paper seems to be deficient. What is the feasible range of R_i,t? How high a R_i,t value indicates a high space-time risk? Is it possible to have a negative or zero R_i,t value, if so, how to express the negative R_i,t value or 0 value?

Response 1: We appreciate your constructive comments. R_i,t is the risk assessment indicator we defined to characterize the spatiotemporal risks of an epidemic. As shown in Equation (3), it denotes the risk assessment value for a certain region i at time t. If a region has a lower level of population density but a higher level of confirmed cases, the risk of epidemic spread exists. Specifically, while a region has a relatively low level of cases but a relatively high level of population density, R_i,t is negative indicates low risk. While the two relative levels of cases and population density are identical, R_i,t equals zero and indicates medium risk. Instead, while the relative level of population density is low but the relative level of is high, R_i,t is positive indicates high risk. In addition, a higher value of R_i,t indicates more risk for region i at time t. Note that this study does not consider the situations of medium and low risk, focusing on the spatiotemporal variations in high-risk areas. In this revised manuscript, we have elaborated on the forementioned narrative (Page 6-7, Lines 236–249).

Point 2: Authors may need to give a clear description or definition of the “space-time risk”, otherwise, readers may possibly not understand this risk index well. For example, the basic reproductive number R_0 has very clear meanings.

Response 2: We appreciate your constructive comments. This work provides a dynamic perspective for identifying the space-time risk of an epidemic in the space-time domain. We examine the disparity of the relative levels of cases compared with population at each date for each specific region across the entire study area. The space-time risk refers to the likelihood of an outbreak around a specific region at a certain date (Page 6, Lines 226–231). Hope the revision can address your concern.

Point 3: The sample data only covers the very initial stage of COVID-19 outbreaks, during which rigorous quarantine measures were taken in China mainland. If longer period datasets can be included, the work will be wonderful.

Response 3: The experimental period in this article is set from January 20, 2020, to April 12, 2020. This period indicates a distinctive phase in the development of the COVID-19 outbreak in China. This study focuses on the 84-day (12-week) period, providing a reasonable empirical basis for our approach. We intend to conduct future work to extend the use of our approach. For instance, incorporating longer period datasets of COVID-19 outbreak in China to enhance the depth and significance of our research, and collecting time-series data of COVID-19 outbreaks in other countries or regions to validate the proposed approach. The forementioned discussion has been included in this revised manuscript (Page 16, Lines 501–504).                                                                                                                                                                                                                                                                                                                                                                                

Point 4: The 12 pictures in figure 8 indicate significant risk variances in space and time. Are these results consistent with the reality? It may be better to include such discussions.

Response 4: Thanks for your constructive comments. The 12 pictures in figure 8 indicate significant risk variances in space and time. In order to validate the relationship between the results of the risk measurement model measuring space-time risk and the reality, in Section 3.4, the relationship between risk values, incidence rates and population migration intensity in six cities, namely, Beijing, Guangzhou, Wuhan, Shanghai, Wenzhou, and Chongqing, is analyzed, and it is found that our proposed method can better describe the space-time risk dynamics in each city (Page 13, Lines 382–386; Lines 409–411). Hope the response can address your concern.

Reviewer 3 Report

Comments and Suggestions for Authors

The authors present an application of the heads/tails aggregation method to COVID-19 data to examine the spatial and temporal structure at the municipal administrative level. 

General Concerns

While the paper is generally successful at integrating COVID-19 data as a specific example of the heads/tails method, I note a few general concerns with the article that the authors should. 

1. As I discuss later under the specific concerns section, the results achieved in this study are predicated on the assertion that the data is aggregated to a 7-day week and collected at the municipal administrative unit. The virus does not adhere or respect either aggregation definition. The authors need to make claim to the fact that a different time period (e.g., 10-day or 14-day) would likely produce different results. Moreover, a different spatial structure (e.g., at a coarser administrative level) would also produce different results. In other words, the results produced by this article are only valid under the assumption of a 7-day week and the municipal administrative district. Other temporal frameworks or enumeration unit boundary structures for the same time period (12 weeks) and geographic area (China) will result in different outcomes. 

2. The authors use the term "risk" quite frequently in this article. In some cases, "risk" is being quantified while in other cases, "risk" is used in more abstract terms (some places have more or less risk than others). For example, "risk" is used in several phrases such as "space-time risk," "epidemic risk assessment," "spatio-temporal risk," "risk dynamics," and so forth. While the difference between these and other "risk" phrases may seem nuanced, these phrases need to be better defined (or "risk" used in a more consistent fashion), especially when "risk" is being used as a quantifying measure.  

3. The map designs need to be re-thought as they are presently diminishing the visualization potential that otherwise could be realized. I have included recommendations on ways to improve the map design to better communicate the mapped patterns. 

In a similar context, the symbolization choice for the map in Figure 1b is incorrect. Choropleth symbolization requires standardized data. The data being mapped in Figure 1b is raw total (number of confirmed cases of COVID-19). 

Specific Concerns

Line 50-51: The "Ten-cent location-based big data" isn't familiar to me.

The choropleth map in Figure 1: The choropleth map is misleading as the vales shown in the administrative boundaries should be standardized to account for variations in the sizes of the administration boundaries. If raw count data is shown, consider using the proportional symbol symbolization method.

Line 195: The case of "ht" is not consistently used. Here, ht is cast in lower case as ht-index. Later, Ht-index is used.

Line 219: Clarify that the ht-index is expressed as the number of recursive segmentations (as stated on the previous page) + 1

Start of Line 223: A quick example would help remove my confusion. The formula expressed in line 223 is a really complex way of simply specifying the position of any segmentation out of the total number of segmentations + 1. Not sure why there is a need to add this complexity to an otherwise simple measure. 

Line 223: The formula doesn't make sense to me. There seems to be a different expression to calculate the ht-index. My confusion rests in the fact that if N is the ht-index (the denominator), then how is the ht-index being calculated if it is already known?

End of line 223: Here, upper case Ht-index is used. Be consistent. There needs to be better distinction between the Ht index and the ht index (both upper and lower case letters are used) - are the Ht and ht indices different measures? This is confusing.

Line 261 & 262: Provide evidence for these statements by referencing Figure 4c - 4d. Otherwise, why include the figures if there is no specific reference to or explanation of their evidence in your discussion. 

Line 266: Why not discuss Figures 4c and 4d here as well?

Figure 4a and 4b: The red-blue dot color on the small plot is really difficult to visually distinguish. I am not sure that adding the smaller two plots for each dataset is really needed. The plots are difficult to see and the data values in the Table are sufficient to show the numerical breaks in the data.

Line 279: Relate back to ht-index value (e.g., does this mean higher ht-index values?). Explain

Line 280: The map design limits this visualization ability - see my recommendations below on improving the map design.

Line 286 & 287: I don't know what you mean by "largest 5 levels?" What is the "stratification structure" (is this the same idea as spatial structure?)? And, "largest" in what context: greatest number of data values, greatest range in data values, etc.? Clarify this point.

Line 288 & 289: The map design in Figure 7 is equally difficult to discern. Please follow my recommendations to improve the map design and thus the readability of the maps.

Line 290: "Notable" how and for what reason? Explain

Line 292: Comment on the importance of this change.

Figure 6: The map design makes visualizing the data symbols and the overall pattern difficult. I have four recommendations on how to better the map design to help visualize the data:

1. Make the administrative boundaries' outline color a light gray rather than black to reduce their visual impact.

2. Replace the red graduated circles with area fills for the administrative boundaries. The fill should employ a sequential hue sequence from light to dark of a color hue to visually emphasize the different hierarchical levels from L1 to L6.

3. There is no need to apply a unique dashed line symbol to the outline of the country. Since there are no surrounding countries to confuse with China's boundaries, there is no need to add visual complexity to the otherwise simple map. 

4. If you change the symbolization method from proportional circles to choropleth symbolization, there won't be a need to show enlarged maps of the area. The size of the proportional circles in the inset maps is not at the same size as shown on the main map - so a different legend is really needed for each inset map. 

Figure 7: See my previous comments on the overall map design as the black administrative boundaries prevent the dark blue circles from readily being visualized.

The dashed box outlines are nearly impossible to see and therefore are invisible to the reader. Why are the dashed lines black rather than blue? Due to this, the reader isn't able to directly link each of these inset maps back to their respective main map.

Line 297: Be consistent how multiple figures are referenced (here "figure" is used twice but later in this caption "figure" is only used once.

Line 303: This is verbose, why not just say "Figure 8 illustrates..."

Line 305: This term is being introduced here for the first time is being associated with risk which is not clearly associated with earlier discussion.

Line 308: What does "single-core" phrase mean?

Line 315: What is the standard deviation of the risk assessment indicator? This is a new metric which hasn't been previously discussed, the reader is uncertain as to this metric or its relevancy to the present statement. Why is this metric important? 

Line 316: and the importance of "increased" is...?

Line 316: If you are speaking in statistical terms, then what were the changes to the p-value? Or, are you more suggesting that this decrease was important rather then significant? Explain

Line 318: To what does "high-risk values" mean...is this prevalence risk, the Ri,t value, or something else? Be consistent with terminology use. I don't know what "high-risk" values mean in this context. 

Figure 8: The map design needs work. I recommend a different color scheme for the area fill colors as the light to dark magenta color values of the presently used scheme are not perceptually logical (the middle magenta values are seen as more visually important than the darker magenta values).

A better legend wording would be "high" and "low" as "strength" and "weakness" doesn't make sense in connection to the discussion.

Figure 9: Some of these metrics need to be explained in better detail and their logic of inclusion be discussed (e.g., standard deviation).

Line 341: What is this "peak risk intensity" metric meaning and is this value derived at all from the present study?

Line 349: Is this relating back to the L value? Explain

Figure 10 caption: Which specific risk value is being reference here? Explain

Line 380 and 381: Is this true that significant research results were achieved or are the results just "important" without being significant

Line 381: A better word choice here would be "valuable" rather than "significant."

Line 297: It is important to point out to the reader that this weekly time-period is arbitrary as the virus and its behavior didn't adhere to the 7-day period that we define as a week; in other words, a different 7-day period may in fact produce different results.

Added to this, a different temporal framework (e.g., 10-days or 14-days) may likely produce different results.

The results obtained in this study are only valid for the time period under consideration; the approach take in this study shows a specific space-time outcome but the specific patterns observed may differ if a different time period is chosen.

In a similar vein, choosing a different spatial structure will likely have different outcomes too. The MAUP should be acknowledged as well. 

Comments on the Quality of English Language

I noted no major issues with the use of the English language in the article. 

Author Response

We sincerely thank you for the appreciation of our work. We have done our best to improve the quality of the manuscript according to the reviewers’ comments. The one-to-one responses are laid out as follows. All the modifications are highlighted in blue in the revised manuscript. Hope the revision can address your concern.

Point 1: General Concerns. While the paper is generally successful at integrating COVID-19 data as a specific example of the heads/tails method, I note a few general concerns with the article that the authors should. As I discuss later under the specific concerns section, the results achieved in this study are predicated on the assertion that the data is aggregated to a 7-day week and collected at the municipal administrative unit. The virus does not adhere or respect either aggregation definition. The authors need to make claim to the fact that a different time period (e.g., 10-day or 14-day) would likely produce different results. Moreover, a different spatial structure (e.g., at a coarser administrative level) would also produce different results. In other words, the results produced by this article are only valid under the assumption of a 7-day week and the municipal administrative district. Other temporal frameworks or enumeration unit boundary structures for the same time period (12 weeks) and geographic area (China) will result in different outcomes.

Response 1: We sincerely appreciate your valuable comments. Indeed, the MAUP effects cannot be ignored. This discussion has been included in the revised manuscript (Page 15, Lines 452–460). Hope the revision can address your concern.

Point 2: The authors use the term "risk" quite frequently in this article. In some cases, "risk" is being quantified while in other cases, "risk" is used in more abstract terms (some places have more or less risk than others). For example, "risk" is used in several phrases such as "space-time risk," "epidemic risk assessment," "spatio-temporal risk," "risk dynamics," and so forth. While the difference between these and other "risk" phrases may seem nuanced, these phrases need to be better defined (or "risk" used in a more consistent fashion), especially when "risk" is being used as a quantifying measure.

Response 2: We appreciate your constructive comments and are sorry for the confusion. This work provides a dynamic perspective for identifying the space-time risk of an epidemic in the space-time domain. We examine the disparity of the relative levels of cases compared with population at each date for each specific region across the entire study area. The space-time risk refers to the likelihood of an outbreak around a specific region at a certain date (Page 6, Lines 226–231). In addition, we use a consistent term “space-time risk” to avoid the misleading in this revised manuscript. Hope the revision can address your concern.

Point 3: The map designs need to be re-thought as they are presently diminishing the visualization potential that otherwise could be realized. I have included recommendations on ways to improve the map design to better communicate the mapped patterns. In a similar context, the symbolization choice for the map in Figure 1b is incorrect. Choropleth symbolization requires standardized data. The data being mapped in Figure 1b is raw total (number of confirmed cases of COVID-19).

Response 3: Thanks for the helpful suggestions. We have revised the thematic mapping (e.g., Page 4, Figure 1b; Page 10, Figure 6).

Point 4: Specific Concerns. Line 50-51: The "Tencent location-based big data" isn't familiar to me.

Response 4: It is location-based big data provided by Tencent Inc. of China. Tencent location-based services represent a prominent application in location services. They acquire 5.7 billion location data points daily, encompassing more than 10% of the global population, with a coverage rate exceeding 70%. Noteworthy for its extensive user base and broad demographic distribution, Tencent location-based services effectively depict individuals’ trajectories and transitions.

Point 5: The choropleth map in Figure 1: The choropleth map is misleading as the vales shown in the administrative boundaries should be standardized to account for variations in the sizes of the administration boundaries. If raw count data is shown, consider using the proportional symbol symbolization method.

Response 5: As mentioned before, we have revised the thematic mapping (Page 4, Figure 1b).

Point 6: Line 195: The case of "ht" is not consistently used. Here, ht is cast in lower case as ht-index. Later, Ht-index is used.

Response 6: We have revised the inconsistent use of this term throughout the main text (e.g., Page 6, Lines 224).

Point 7: Line 219: Clarify that the ht-index is expressed as the number of recursive segmentations (as stated on the previous page) + 1

Response 7: Thanks for your helpful suggestion. In this revised manuscript, we show that the ht-index is represented as the number of recursive segments (as described on the previous page) + 1 (Page 6, Lines 218-220).

Point 8: Start of Line 223: A quick example would help remove my confusion. The formula expressed in line 223 is a really complex way of simply specifying the position of any segmentation out of the total number of segmentations + 1. Not sure why there is a need to add this complexity to an otherwise simple measure.

Response 8: Sorry for the confusion. We have revised the narrative. Li/N is the hierarchical level of each city in the total number of levels, i.e., Li/N denotes the ht-index rank, where Li represents hierarchical structure levels, and N is the ht-index (Page 6, Lines 222-224). This lays the theoretical foundation for the subsequent risk measurement model.

Point 9: Line 223: The formula doesn't make sense to me. There seems to be a different expression to calculate the ht-index. My confusion rests in the fact that if N is the ht-index (the denominator), then how is the ht-index being calculated if it is already known?

Response 9: As mentioned earlier, this is expressed to further illustrate the hierarchical level of each city in the total number of levels. Li/N is the hierarchical level of each city in the total number of levels, i.e., Li/N denotes the ht-index rank, where Li represents hierarchical structure levels, and N is the ht-index (Page 6, Lines 222-224). This lays the theoretical foundation for the subsequent risk measurement model.

Point 10: End of line 223: Here, upper case Ht-index is used. Be consistent. There needs to be better distinction between the Ht index and the ht index (both upper and lower case letters are used) - are the Ht and ht indices different measures? This is confusing.

Response 10: As mentioned earlier, we have revised the inconsistent use of this term throughout the main text. The consistent term of “ht-index” is used in this revised manuscript (e.g., Page 6, Lines 224).

Point 11: Line 261 & 262: Provide evidence for these statements by referencing Figure 4c - 4d. Otherwise, why include the figures if there is no specific reference to or explanation of their evidence in your discussion.

Response 11: Thanks for your helpful suggestion. In this revised manuscript, we have provided some additional information (Page 7, Lines 275-288).

Point 12: Line 266: Why not discuss Figures 4c and 4d here as well?

Response 12: As mentioned earlier, we have added a discussion of Figures 4c and 4d to the revised manuscript (Page 7, Lines 275-288).

Point 13: Figure 4a and 4b: The red-blue dot color on the small plot is really difficult to visually distinguish. I am not sure that adding the smaller two plots for each dataset is really needed. The plots are difficult to see and the data values in the Table are sufficient to show the numerical breaks in the data.

Response 13: We appreciate for your constructive comments. We have revised the information revealed by Figures 4a and 4b (Page 7, Lines 259-271). For example, in Figure 4a, the largest plot encompasses population density data for all 367 cities. Dark red indicates the first head (126 cities), while dark blue represents the first tail (241 cities). Replotting the population density rank-size plot for the 126 cities in the first head reveals that dark red denotes 30 cities with population density in the head, while dark blue represents 96 cities with population density in the tail. The smallest plot is for the population density rank-size plot within the second head, consisting of 30 cities, and it exhibits heavy-tailed characteristics in population density data (Page 7, Lines 265-271).

Point 14: Line 279: Relate back to ht-index value (e.g., does this mean higher ht-index values?). Explain

Response 14: The ht-index indicates the spatial hierarchy, e.g., population density ht-index=6 means that there are 6 hierarchical levels. In the revised manuscript, we have further explained this issue. (Page 8-9, Lines 306-314).

Point 15: Line 280: The map design limits this visualization ability - see my recommendations below on improving the map design.

Response 15: We have provided some additional information in this revised manuscript (Page 10, Figure 6).

Point 16: Line 286 & 287: I don't know what you mean by "largest 5 levels?" What is the "stratification structure" (is this the same idea as spatial structure?)? And, "largest" in what context: greatest number of data values, greatest range in data values, etc.? Clarify this point.

Response 16: Sorry for the confusion. We have revised the narrative (Page 9, Lines 322-324).

Point 17: Line 288 & 289: The map design in Figure 7 is equally difficult to discern. Please follow my recommendations to improve the map design and thus the readability of the maps.

Response 17: Thanks for your suggestions. we have revised the thematic mapping (Page 10, Figure 6).

Point 18: Line 290: "Notable" how and for what reason? Explain

Response 18: Sorry for the confusion. We have revised the interpretation in this revised manuscript (Page 9, Lines 328-331). Hope the revision can address your concern.

Point 19: Line 292: Comment on the importance of this change.

Response 19: In our revised manuscript, we have interpreted the importance of this change. It indicates that ht-index plays an important role in revealing the hierarchical structure inherent in the space of things, and it can portray the change of the level in which the new crown pneumonia cases are in each city (Page 9, Lines 333-337).

Point 20: Figure 6: The map design makes visualizing the data symbols and the overall pattern difficult. I have four recommendations on how to better the map design to help visualize the data: 1. Make the administrative boundaries' outline color a light gray rather than black to reduce their visual impact. 2. Replace the red graduated circles with area fills for the administrative boundaries. The fill should employ a sequential hue sequence from light to dark of a color hue to visually emphasize the different hierarchical levels from L1 to L6. 3. There is no need to apply a unique dashed line symbol to the outline of the country. Since there are no surrounding countries to confuse with China's boundaries, there is no need to add visual complexity to the otherwise simple map. 4. If you change the symbolization method from proportional circles to choropleth symbolization, there won't be a need to show enlarged maps of the area. The size of the proportional circles in the inset maps is not at the same size as shown on the main map - so a different legend is really needed for each inset map.

Response 20: Thank you for your suggestions regarding our mapping and visualization aspects. We have made the changes in response to the four suggestions you gave us in this revised manuscript (Page 10, Figure 6). Specifically, the outline color of administrative boundaries has been changed to light gray to minimize visual impact. We have employed a gradient fill, ranging from light to dark tones, to emphasize the different hierarchical levels from L1 to L6 within administrative unit areas. The dashed representation of national borders has been omitted to reduce complexity. The graphic has been changed the symbolization method from proportional circles to choropleth symbolization

Point 21: Figure 7: See my previous comments on the overall map design as the black administrative boundaries prevent the dark blue circles from readily being visualized.

Response 21: As mentioned earlier, we have revised the thematic mapping (Page 10, Figure 6).

Point 22: The dashed box outlines are nearly impossible to see and therefore are invisible to the reader. Why are the dashed lines black rather than blue? Due to this, the reader isn't able to directly link each of these inset maps back to their respective main map.

Response 22: As mentioned earlier, we have revised the thematic mapping (Page 10, Figure 6).

Point 23: Line 297: Be consistent how multiple figures are referenced (here "figure" is used twice but later in this caption "figure" is only used once.

Response 23: Thanks for the helpful suggestions. In this revised manuscript, we have made changes in response to your suggestions for consistency (Page 10, Lines 341-342).

Point 24: Line 303: This is verbose, why not just say "Figure 8 illustrates..."

Response 24: We have revised it (Page 10, Lines 346).

Point 25: Line 305: This term is being introduced here for the first time is being associated with risk which is not clearly associated with earlier discussion.

Response 25: We appreciate for your constructive comments and are sorry for the confusion. In this revised manuscript, we corrected "prevalence risk" to "space-time risks". Space-time risk refers to the likelihood of an outbreak in a specific space at a particular time. In this study, we utilize R_i,t to characterize the space-time risk of COVID-19, as elaborated in Section 2.2.3 for detailed information (Page 6, Lines 229-230).

Point 26: Line 308: What does "single-core" phrase mean?

Response 26: The term "single core" refers to a distribution pattern with China as the reference scale, where high-risk cities are concentrated in Wuhan and its surrounding areas, forming a single-core distribution pattern centered around Wuhan, as illustrated in Figure 8d (Page 10, Lines 350-351).

Point 27: Line 315: What is the standard deviation of the risk assessment indicator? This is a new metric which hasn't been previously discussed, the reader is uncertain as to this metric or its relevancy to the present statement. Why is this metric important?

Response 27: To delineate more detailed information about the risk, we computed the standard deviations of risk estimates for Week 1 to Week 12. In the revised manuscript, we have included a more detailed description of the standard deviation (Page 11, Lines 359-365).

Point 28: Line 316: and the importance of "increased" is...?

Response 28: The average value and standard deviation of the risk assessment indicator noticeably increased show that the implementation of policies promoting the active resumption of economic activities has, to a certain extent, exacerbated the risk of the pandemic. Although the number of cities with elevated risk decreased significantly during the study period, it is crucial to pay closer attention to the later stages of epidemic control, especially in cities with extremely high-risk values (Page 11, Lines 371-374).

Point 29: Line 316: If you are speaking in statistical terms, then what were the changes to the p-value? Or, are you more suggesting that this decrease was important rather than significant? Explain

Response 29: Here, we primarily investigate the changes in the number of cities at risk from Week 1 to Week 12, along with the statistical metrics of the average, standard deviation, and maximum risk values per week, aiming to delineate more detailed information about the risk (Page 11, Lines 359-367). Therefore, it is unrelated to p-values. As indicated in Figure 8b-c (Page 13), during the later four weeks (Week 9-Week 12), both the mean and standard deviation exhibit a noticeable decreasing trend over time (Page 11, Lines 369-371). Hope the response can address your concern.

Point 30: Line 318: To what does "high-risk values" mean...is this prevalence risk, the Ri,t value, or something else? Be consistent with terminology use. I don't know what "high-risk" values mean in this context.

Response 30: As mentioned earlier, space-time risk refers to the likelihood of an outbreak in a specific space at a particular time. In this study, we use R_i,t to characterize the space-time risk of COVID-19, and a higher value of R_i,t indicates increased risk, as elaborated in Section 2.2.3 for detailed information(Page 6, Lines 229-246).

Point 31: Figure 8: The map design needs work. I recommend a different color scheme for the area fill colors as the light to dark magenta color values of the presently used scheme are not perceptually logical (the middle magenta values are seen as more visually important than the darker magenta values).

Response 31: As mentioned earlier, we have revised the thematic mapping (Page 12, Figure 7).

Point 32: A better legend wording would be "high" and "low" as "strength" and "weakness" doesn't make sense in connection to the discussion.

Response 32: Thanks for your suggestions. We have revised it (Page 12, Figure 7).

Point 33: Figure 9: Some of these metrics need to be explained in better detail and their logic of inclusion be discussed (e.g., standard deviation).

Response 33: Thanks for the constructive comments. In this revised manuscript, we have provided a detailed explanation on these indicators and discussed their meanings (Page 11, Lines 362-365).

Point 34: Line 341: What is this "peak risk intensity" metric meaning and is this value derived at all from the present study?

Response 34: Sorry for the confusion. There was an issue with the expression here, and we have made the following revision: the risk intensity during this period reached a high of 0.42 (Page 13, Line 396).

Point 35: Line 349: Is this relating back to the L value? Explain

Response 35: As mentioned earlier, this is related to ht-index rank (Page 6, Line 223).

Point 36: Figure 10 caption: Which specific risk value is being reference here? Explain

Response 36: Figure 10 caption refers to the specific value of R_i,t. As mentioned earlier, space-time risk refers to the likelihood of an outbreak in a specific space at a particular time. In this study, we use R_i,t to characterize the space-time risk of COVID-19 (Page 6, Lines 229–230).

Point 37: Line 380 and 381: Is this true that significant research results were achieved or are the results just "important" without being significant?

Response 37: Thanks for your comments. We have revised the narrative (Page 15, Line 435).

Point 38: Line 381: A better word choice here would be "valuable" rather than "significant."

Response 38: We have revised the narrative (Page 15, Line 436).

Point 39: Line 297: It is important to point out to the reader that this weekly time-period is arbitrary as the virus and its behavior didn't adhere to the 7-day period that we define as a week; in other words, a different 7-day period may in fact produce different results.

Response 39: We appreciate your comments and have included this discussion in this revised manuscript (Page 15, Line 452-455).

Point 40: Added to this, a different temporal framework (e.g., 10-days or 14-days) may likely produce different results.

Response 40: As mentioned above, we have included this discussion (Page 15, Line 452-459).

Point 41: The results obtained in this study are only valid for the time period under consideration; the approach take in this study shows a specific space-time outcome but the specific patterns observed may differ if a different time period is chosen.

Response 41: As mentioned earlier, we have included the corresponding explanations in the discussion section (Page 15, Line 452-459).

Point 42: In a similar vein, choosing a different spatial structure will likely have different outcomes too. The MAUP should be acknowledged as well.

Response 42: As mentioned earlier, in this revised manuscript, we have included the corresponding explanations in the discussion section (Page 15, Line 459-460). Hope the revision can address your concern.

Round 2

Reviewer 1 Report

Comments and Suggestions for Authors

The authors adopted my remarks from the previous review.

They added a comparison with another approach from the literature. 

Also, they described the scientific contribution of their work.

My opinion is that paper is well and usefull study and is recommended for acceptance.

Author Response

Point 1: The authors adopted my remarks from the previous review. They added a comparison with another approach from the literature. Also, they described the scientific contribution of their work. My opinion is that paper is well and useful study and is recommended for acceptance.

Response 1: We sincerely thank you for the appreciation of our revision.

Reviewer 2 Report

Comments and Suggestions for Authors

Why the highest value is 0.84?

The high-risk is classified into three levels: low-high, mid-high, and high-high which only appear in the explanation of R_it (lines 248-249). They also can be used to specify the results.

Comments on the Quality of English Language

 Minor editing of English language required

Author Response

Point 1: Why the highest value is 0.84?

Response 1: The positive value of the risk assessment indicator represents the epidemic risk across the space-time domain. Its greater value indicates the higher risk. Specifically, while a region exhibits the highest hierarchical level of cases at a certain date, and meanwhile exhibits the lowest level of population density, the epidemic risk reaches the highest (approaching 1). According to Equation (3), the highest risk value was calculated up to 0.84. In this revised manuscript, we have revised the narrative (Page 7, Lines 248–253). Hope the revision can address your concern.

Point 2: The high-risk is classified into three levels: low-high, mid-high, and high-high which only appear in the explanation of R_it (lines 248-249). They also can be used to specify the results.

Response 2: Thanks for your pointing out the lack. We are sorry for the confusion. This is a carelessness in our previous revision. In this revised manuscript, we do not utilize this risk classification to conduct the thematic mapping. We have deleted this sentence in this revision (Page 7, Lines 248–253).